# Risk Factors for Stunting among Children under Five Years in the Province of East Nusa Tenggara (NTT), Indonesia

**DOI:** 10.3390/ijerph20021640

**Published:** 2023-01-16

**Authors:** Made Ayu Lely Suratri, Gurendro Putro, Basuki Rachmat, Noer Endah Pracoyo, Aris Yulianto, Anton Suryatma, Mohamad Samsudin

**Affiliations:** Center for Public Health and Nutrition Research, Health Research Organization, National Research and Innovation Agency, Jakarta 10340, Indonesia

**Keywords:** stunting factors, child malnutrition, East Nusa Tenggara province, child growth stunting

## Abstract

In East Nusa Tenggara Province, Indonesia, 42.6% of children under five had stunted growth in 2018, which affects both individual and communal levels. The first step in creating effective interventions is identifying the risk factors for stunting. This study aims to pinpoint the stunting risk factors in East Nusa Tenggara Province, Indonesia, by incorporating secondary data from the 2018 Indonesia Basic Health Research (RISKESDAS). This study implemented a cross-sectional design by utilizing the data of individuals who were successfully visited during the survey. Initial data screening in East Nusa Tenggara Province based on the criteria for children aged 0–59 months and stunting showed as many as 1643. Multivariate logistic regression analysis was performed to evaluate the relationship between children’s characteristics and stunting. There was a significant relationship between age group variables for younger children (aged 12–23, 24–35, and 36–47 months), mothers with low education, and children living in rural areas with the incidence of stunting in children (*p*-value < 0.05). The dominant factors that caused stunting in this study were the children’s age of 24–35 months (OR = 2.08, 95% CI: 1.12–3.86), mothers with low education (OR = 1.57, 95% CI: 1.18–2.08), and children living in rural areas (OR = 1.39, 95% CI: 1.01–1.91). The highest prevalence of stunting was in the group of children aged 12–23 months (45.2%). To prevent child stunting, the government must intervene for mothers with low education and those living in rural areas. Intervention includes intensive socialization about improving nutritional status during pregnancy and practicing complementary feeding and breastfeeding habits until the child is 24 months old.

## 1. Introduction

Chronic nutritional problems are caused by a lack of nutrient intake in the long term, resulting in unfulfilled nutritional needs and causing stunting [1,2]. In comparison to other middle-income countries, Indonesia is one of the countries with a high prevalence of stunting [3]. In comparison with the ASEAN nations, Indonesia still has a lower frequency of stunting than Myanmar (35%), but it is still greater than Vietnam (23%), Malaysia (17%), Thailand (16%), and Singapore (4%) [4]. One of the global problems impeding human growth is stunting, which is present in children with low height-for-age due to chronic malnutrition [5]. Stunting, which can occur from pregnancy until 24 months, is a type of growth failure resulting from a long-term accumulation of inadequate nutrition [6]. Short-term consequences of malnutrition include increased morbidity and mortality, developmental disorders (cognitive, motoric, language), and increased economic burden for the cost of care and treatment of sick children [7]. In the long term, it causes a decline in reproductive health, learning concentration, and low work productivity [8]. In a larger context, short children growing up to be adults with low education levels, poverty, poor health, and susceptibility to non-communicable diseases—such as obesity, cardiovascular disease, etc., will further impact state losses [9].

Several studies show that stunting poses risks of a decrease in academic achievement [10], increased risk of obesity [11], susceptibility to non-communicable diseases [12], and an increased risk of degenerative diseases [10,13]. Stunting, wasting, and underweight are indicators of malnutrition in children. Stunting is the term used to describe a child who has a low height for their age, wasting is used to describe a child who is too thin for their height, and underweight is used to describe a child who has low weight for their age [14]. The 2006 WHO child growth standards are recommended for use in international contexts to construct height-for-age, weight-for-height, and weight-for-age z-scores [15]. If a child’s height-for-age z-score (HAZ) is less than two standard deviations compared to the WHO Child Growth Standards median of the same age and sex, they are called stunted. Weight for height z-score (WHZ) below two standard deviations is considered wasted and indicates acute undernutrition or fast weight loss. Weight-for-age z-score (WAZ) under two standard deviations is considered underweight [16].

The WHO has declared the resolution of global targets on maternal and child nutrition as a priority [17]. The primary goal is to reduce childhood stunting by 40% globally or 3.9% between 2012 and 2025 [18]. In addition, the Indonesian government has issued several policies and regulations to address the malnutrition problem, particularly stunting [19]. Implementing these policies and regulations is hoped to assist the management of stunting through monitoring and evaluation activities that play a very strategic and crucial role in ensuring the impact of interventions on stunting prevention and control, which further contributes to the reduction of the stunting problem in Indonesia.

According to 2018 RISKESDAS, the proportion of stunted children under five in Indonesia has dropped from 37.2% to 30.8% [20,21]. The 2018 RISKESDAS data showed that East Nusa Tenggara province has the highest stunting rate in Indonesia, with a stunting prevalence of 42.6%, which declined from the previous survey (51.7%). The causes of stunting in Indonesia can occur due to various factors, such as the inappropriate practice of complementary feeding, exposure to viruses, poor breastfeeding habits, and inadequate maternal nutrition. Moreover, some distal determinants such as poor water quality and sanitation (services and infrastructure), health care services, food systems, and education have also affected stunting prevalence [22]. Based on this data, some variables expose the population to the factors that cause child stunting, and the most vulnerable people must be the focus of intervention. Therefore, the first step in creating effective interventions is identifying the risk factors for stunting. This study aims to pinpoint the stunting risk factors in East Nusa Tenggara province, Indonesia.

## 2. Methodology

### 2.1. Data Collection and Research Design

The data used for analysis in this research were secondary data from the 2018 RISKESDAS—a nationally representative cross-sectional survey conducted in 34 provinces in Indonesia from April to May 2018. All Indonesian households served as the research population. A stratified multistage systematic random sampling and the probability proportional to size (PPS) approach was performed to collect the research sample from selected households. In total, 295,720 households were successfully visited, with 1,091,528 people living in those households [23].

Enumerator training was conducted to train the use of the research questionnaires and apply certain communication strategies before data collection began. Village representatives and health professionals accompanied the enumerators as they visited residences to collect data. The enumerators requested that all chosen family members read and sign the informed consent form before the interviews began. A paper questionnaire with structured questions was used as a research instrument for data collection. The research questionnaire was divided into two sections: the first section consists of household-related questions, and the second section asks about individuals. The anthropometric measurement included obtaining children’s body weight using a digital scale with an accuracy of 0.1 kg, and body height measurement using a height measuring instrument with an accuracy of 0.1 cm. After collecting the data, the team leader and the enumerator reviewed the completed questionnaire. Afterward, enumerators entered the data from the questionnaire into the Census and Survey Processing System’s data entry program (CSPro) version 7.3 [24].

This research was a follow-up analysis of the 2018 RISKESDAS data for East Nusa Tenggara province. Cleaned and selected data for East Nusa Tenggara province consists of 22 districts/cities, 1088 census blocks, and 10,880 households. A total of 1643 data was obtained from a successfully visited individual—which, in the current research context, means children under five in selected households [20].

### 2.2. Dependent Variables

The dependent variable in this study uses the nutritional status of toddlers (stunting). A child’s height achieved at a certain age is used as an indicator of nutritional status, one indicator of which is stunting. According to the WHO, growth standards can be assessed using a z-score value, which represents an indicator of height according to age or height deviation from the average height. The nutritional status category limits for toddlers based on the height-for-age index in this study consisted of two categories: stunting (z-score < −2.0 SD) and normal (z-score ≥ −2.0 SD).

The z-score value in the 2018 RISKESDAS data was determined by measuring age, weight, and height as a basis for measuring children’s nutritional status. Age, weight, and height variables were used to measure the children’s nutritional status. The children’s weight (W) and height (H) variables in the data were in the form of three anthropometric indicators; weight-for-age (W/A), height-for-age (H/A), and weight-for-height (W/H). In addition, the body mass index (BMI) formula was used to calculate the nutritional status of children under five. Furthermore, the weight and height of each child were converted to a standard value (z-score) using the WHO anthropometric standard for children [25]. The z-score value of each indicator was then used to determine the nutritional status of children under five. The nutritional status classification to determine stunting under five years was based on the height-for-age z-score (HAZ) indicator. 

### 2.3. Independent Variables

Independent variables consist of data on the child’s sex under five, low birth weight (LBW), supplementary feeding, monitoring and development, place of residence, access to health care facilities, age, education level, and occupation of the children’s mother in East Nusa Tenggara province, Indonesia. The education level of the children’s mother was divided into two categories, graduating from Junior High School or lower education (≤JHS) and graduating from Senior High School or higher education (>SHS). According to the World Health Organization (WHO), low birth weight is a birth weight of less than 2500 g (up to and including 2499 g) [26]. 

Supplementary feeding for toddlers is nutritional supplementation in the form of additional food with unique formulations and fortified with vitamins and minerals with the target of toddler groups for recovery or fulfillment of nutritional status [27]. Supplemental food for toddlers in the 2018 RISKESDAS questionnaire refers to all food given during Integrated Healthcare Center activities (Posyandu during supplementary feeding counseling). Provision of given supplemental food for underweight toddlers for 90 days of eating (supplementary feeding recovery). Supplementary feeding recovery, namely, complementary food for ASI in the form of biscuits containing ten vitamins and seven minerals. For supplementary feeding recovery in children aged 12–24 months, who use biscuits with nutritional value: total energy 180 kcal, 6 g fat, and 3 g protein. The serving size contained 29 g of total carbohydrates, 2 g of dietary fiber, 8 g of sugar, and 120 mg of sodium. Meanwhile, there are two types of supplementary feeding for recovery based on local food ingredients, namely weaning foods for infants and children aged 6–23 months and additional food for recovery for toddlers 24–59 months in the form of family food. Other than that, supplemental food was also obtained through assistance from other parties, such as donations from NGOs/companies or certain parties doing certain activities or promoting specific products [20].

Access to healthcare facilities was measured by an indicator of knowledge, using several questions at the household level. The indicators were analyzed using the Principal Component Analysis (PCA) method, which consists of three dimensions: the costs incurred for round-trip transportation to the healthcare facility, the round-trip time from home to the healthcare facility, and the type of transportation used to reach the healthcare facility. PCA was used to simplify several variables into one meaningful variable. The 2018 RISKESDAS has three types of access to health services: access to hospital facilities, access to public health center (Puskesmas) facilities, and access to independent clinic/practice facilities. The index of access to health facilities was divided into two categories: (1) easy and (2) difficult [20].

### 2.4. Statistical Analysis

The frequency distribution of the independent and dependent variables was included in the univariate analysis. Bivariate analysis determines the association between independent and dependent variables. Meanwhile, a multivariate analysis was performed with a complex sample approach to see the effect of the independent variables on the incidence of stunting. Multivariate logistic regression analysis aimed to study the relationship of several independent variables with one dependent variable simultaneously. This research considers strata values, primary sample units, and the average of weighted variables because of a complex survey design. The number of weighted N samples was the number of samples produced via complex sample analysis. The multivariate logistic regression model included variables in the bivariate model with *p*-values less than 0.25 [28]. The final model used in this research had all factors significantly associated with stunting, using a significance threshold of 0.05 as the cut-off. The odds ratio (OR) and confidence interval (CI) of 95% for each variable in the final model was displayed. The study’s regression equation was as follows:(1)$$\ln\left(\frac{\hat{p}}{1-\hat{p}}\right)=\beta_{0}+\beta_{1}X$$
wherein: Natural Logarithm, where $\beta_{0}+\beta_{1}X$ is an equation commonly known in OLS, while $\hat{p}$ is the logistic probability obtained from the following formula:(2)$$\hat{p}=\frac{\exp\left(\beta_{0}+\beta_{1}X\right)}{1+\exp\left(\beta_{0}+\beta_{1}X\right)}=\frac{e^{\left(\beta_{0}+\beta_{1}X\right)}}{1+e^{\left(\beta_{0}+\beta_{1}X\right)}}$$

### 2.5. Ethical Clearance

The National Institute of Health Research and Development, Ministry of Health, Republic of Indonesia, granted ethical approval for this research on 24 January 2018, with the reference number LB.02.01/2/KE.024/2018.

## 3. Results

### 3.1. Descriptive Analysis

There were 1643 children under five involved as research respondents, which consisted of 838 males and 805 females. Table 1 shows that the majority of respondents were in the age group of 48–59 months (23.07%). Of the total 1643 respondents, 636 children under five (38.71%) experienced stunting. Some children had received a supplementary feeding program (30.07%) from the local health post. Children under five who received monitoring and development programs at the integrated health care center, represented as many as 1345 people (81.86%). Moreover, the mothers of children under five years with low education were 865 (52.65%), while the mothers who worked were 1080 (65.73%). Most children under five years (*n* = 7174, 71.46%) lived in rural areas, with the majority of children having the most difficult access to healthcare facilities (*n* = 1128, 68.66%).

### 3.2. Bivariate Analysis

Table 2 shows the bivariate test results, which prove the relationship between the independent and dependent variables. Based on age group, children aged 12–23 months, 24–35 months, and 36–47 months were significantly associated with stunting (*p*-value < 0.05). The lower the age group of children under five, the more likely they were to be at risk of stunting. According to education level, mothers with low education were significantly associated with stunting in children under five (OR = 1.72, 95% CI: 1.31–2.25; *p*-value < 0.05). In terms of place of residence, children living in rural areas are significantly associated with stunting (OR = 1.47, 95% CI: 1.1–1.99; *p*-value < 0.05). However, the variables of the child’s sex, receiving supplemental feeding, mother’s age, mother’s occupation, monitoring and development of children, and access to healthcare facilities were not significantly associated with the prevalence of stunting in children under five (*p*-value ≤ 0.05).

### 3.3. Multivariate Analysis

After the multivariate logistic regression test was performed, the results of which are shown in Table 3, we found three variables—the child’s age, mother’s education, and child’s place of residence, significantly related to the incidence of stunting in children under five, with a *p*-value of 0.05. Children aged 24–35 months were 2.08 times more likely to experience stunting than children aged 0–11 months (OR = 2.08, 95% CI: 1.12–3.86). Mothers with low education had a 1.57 times risk of having stunted children compared to mothers with higher education (OR = 1.57, 95% CI: 1.18–2.08). Moreover, mothers who lived in rural areas were at higher risk of having stunted children by 1.39 times compared to mothers who lived in urban areas (OR = 1.39, 95% CI: 1.01–1.91).

## 4. Discussion

Male and female child proportions in this study were comparable. The findings of this research prove that there was no relationship between the child’s sex and the incidence of stunting in children under five (OR = 0.98; 95% CI:0.74–1.29). This finding supports several research results that there was no relationship between age and the incidence of stunting in children [29,30,31,32]. According to the research, boys (37.8%) were more likely to be stunted than girls (37.3%). However, according to the research findings, the prevalence rate of stunting in boys (37.8%) was slightly higher than for girls (37.3%). This finding was in line with Utami RA et al.’s research, which stated that the prevalence of stunting was lower in girls (37.8%) than in boys (48.8%). Other studies have found that statistically, the child’s sex difference in stunting prevalence is negligible for most age groups. According to the research by Zhang et al., there was no child’s sex difference in the risk of stunting between boys and girls [33]. Another possibility is that boys may experience stunting because boys are more susceptible to infections and other diseases that might disrupt their normal growth [8].

The most dominant factor which caused stunting in this research was the child’s age. The highest prevalence of stunting was in the group of children aged 12–23 months (45.2%). The analysis results at the age limit of 12–59 months showed that the lower the age group of the children, the more likely they were to be at risk of stunting. However, the risk of stunting was lower in children aged 0–11 months. Research findings in South Africa showed similar results to this study, where stunting was more likely to be experienced by children aged 12–23 months than children aged 11 months [34]. Analysis at the national level using the same data sources reported that stunting was more likely to be experienced by children aged 12–23 months than by children aged 11 months [35]. This finding contrasts with several studies which stated that younger children (6–23 months) had a reduced risk of stunting compared to older children (24–59 months) [8,36]. East Nusa Tenggara Province has a high percentage of exclusive breastfeeding for the first six months (97.41%), which may have a protective effect against stunting at a young age [23]. Inappropriate complementary feeding practices during the weaning stage, when infants switch from exclusive breastfeeding to including complementary foods in their diet, can lead to increased stunting prevalence among children under five in East Nusa Tenggara [37,38]. If children are only given continued breastfeeding without being given complementary foods at the right age, problems with the child’s growth will arise. Poor linear growth may result if a child obtains insufficient supplemental feeding in response to the increased nutritional demand [39,40]. In addition, as children become older, they are exposed to more childhood illnesses and disorders, including food with poor hygiene and unclean environments, which may disrupt their growth [41].

Providing additional food for a child to meet their nutritional needs is essential, especially for those suffering from stunting. Supplemental feeding is giving food in the form of biscuits guaranteed to be safe and of good quality and paying attention to aspects of the nutritional value needed by the targeted child [42]. Supplementary feeding can be in the form of products rich in nutrients and healthy foods. Insufficient supplemental feeding produces insufficient energy, protein, and micronutrient intake, which impacts growth. This study showed no relationship between supplementary feeding and child stunting (*p*-value = 0.272). However, the research results by Kevin H. Hosang et al. showed that supplementary feeding significantly effected changes in the nutritional status of children under five with malnutrition in public health centers in Manado City, Indonesia [43]. In line with previous research, Safrina et al. stated that there was a relationship between supplementary feeding and the risk of stunting in children under five years [44]. However, a recent meta-analysis showed that for studies included in the meta-analysis with low certainty evidence, there was no evidence of an effect or an unclear effect of nutrient supplementation of children on HFA (height-for-age). For studies published in a narrative style with very low to moderate certainty evidence, there was no evidence of an effect on length [37].

The second dominant factor which caused stunting in this study was the education level of the child’s mother. Children of mothers with low education were at a risk of stunting 1.6 times higher than those of mothers with higher education. These results were consistent with the findings of several studies, which stated that there was a relationship between maternal education level and the prevalence of stunting [45,46]. A mother’s education affects the pattern of parenting, including nutrition regulation during feeding practices and maintaining children’s health [47,48,49]. Educated mothers can receive information from outside and increase their understanding and knowledge about child-rearing. Therefore, education is related to the mother’s knowledge of how to apply parenting patterns and the nutritional status of children. According to research done by Hagos et al., maternal education reduces the risk of stunting and severe stunting [50]. According to their research, children under five whose mothers had not completed elementary or higher education had a 20% higher risk of stunting than those whose mothers had high education levels. According to Torlesse H. et al. (2016), compared to children whose mothers had completed high school (23.0%), the frequency of stunting was higher among children whose mothers had not completed elementary education (43.4%) or had completed primary education (31.0%) [51]. Recently, research results proved that stunting independently relates to social disadvantage. Children with social, economic, political, and emotional (SEPE) disadvantages are shorter than children from affluent backgrounds [52]. Other than that, girls from poor schools were more affected by parental education, development, and individual school ratings than were males from public and private schools. This emphasizes parental education’s importance, especially for girls from the most disadvantaged socioeconomic backgrounds. The third dominant factor of stunting in this research was the place of residence. Mothers living in rural areas were 1.4 times more at risk of having stunted children than mothers living in urban areas. According to the data from the Indonesia Central Statistics Bureau, from 2016–2020, the number of poor people in East Nusa Tenggara in rural areas did not experience a significant decline in 5 years, with a range of 25.46–25.26%, while the number of poor people in urban areas decreased from 11.28% to 8.76% [53]. In addition, the per capita income per month of people in the rural area was under the poverty line in five respective years (2016–2020) and did not experience a significant increase (range IDR 281,002–IDR 377,246). This figure is still below the per capita income per month of rural communities at the poverty line at the national level of IDR 350,420–IDR 437,902 [54]. According to research from Nepal, Peru, and Cambodia, educated mothers from affluent households were more likely to be aware of their children’s nutritional needs [55,56,57]. The purchasing capacity for food and other nutritional items necessary for children’s health is proxied by a household’s wealth. Stunting in children under five will therefore be more common among families with low socioeconomic status since they are less likely to receive a healthy diet [8]. The research results by Gebru, KF. et al. show that children whose parents live in rural areas have a higher chance of experiencing stunting than those who live in urban areas due to a complete healthcare system and easier access to healthcare facilities. Moreover, urban populations usually have higher education and economic status [58]. However, according to research in Peru, child stunting rate has been linked to living in the rural highlands. Environmental and climatic reasons, dietary variations caused by food availability in the highlands versus the lowlands, and the higher levels of physical activity for children living in the highlands may play a significant role [59].

Meanwhile, the child’s sex, supplementary feeding, mothers’ age, occupation, and access to health care facilities were not significant factors for stunting in East Nusa Tenggara province from the results of data analysis. However, according to a review of the literature at the national level with reliable data, nonexclusive breastfeeding for the first six months, low household socioeconomic status, early birth, short birth length, and low maternal height and education all contribute to child stunting in Indonesia [22]. Research in Malawi, involving 840 healthy children aged 6 months in rural Malawi using a randomized assessor-blinded trial method based on exploratory analyses, indicates that giving at-risk children milk-LNS, but not soy-LNS, encourages linear growth, especially between 9 and 12 months [60]. Unfortunately, in this population, the intervention does not seem to be enough to encourage catch-up growth or maintain normal growth after the first year of life. This may be due to the population’s increasing nutrient needs [61], high rates and prevalence of morbidity [62], environmental enteropathy [63], programming of the postnatal growth rate as a result of in utero malnutrition [64], and SEPE backwardness [52].

Furthermore, the analysis results showed no relationship between monitoring and development of children under five at the Integrated Healthcare Center (Posyandu) with the incidence of stunting. Utilization of Posyandu through weighing, measuring body length, providing various information about maternal and child health (breastfeeding, solid food, and disease prevention), and monitoring children’s growth and development are efforts that have been made to prevent children from the risk of malnutrition. However, a meta-analysis suggests more research is required to determine the effects of multi-sectoral interventions that combine nutrition-sensitive and specific methods and programs, as well as the impact of ‘upstream’ practices and policies of governmental, non-governmental, and private sector businesses on nutrition-related outcomes such as stunting [37].

### Study Strengths and Limitation

The strength of this study was that the 2018 RISKESDAS data uses a large sample size and nationally representative data, used to achieve the objective of investigating the relationship between various factors related to stunting in children 0–59 months in the East Nusa Tenggara province. In addition, multivariate logistic regression modeling using sample weights in our analysis can also help minimize bias, thereby enabling an investigation of the influence of provinces, districts, and clusters on stunting [20]. However, the first limitation of this study is that the 2018 RISKESDAS data were implemented in a cross-sectional design that only reported the conclusion of the relationship and did not allow to conclude causality. The second limitation is that the quality of data on stunting in children depends on the memory ability of the mother in providing relevant information. The third limitation is the limited availability of data on the variables displayed in the article. The article cannot display other factors that are known to be associated with stunting, such as quintile values (showing differences in household wealth levels) [65], infections and diseases [66], short maternal stature [41,67], information on feeding, mean of triceps and subscapular skinfold ($\bar{x}$SF) [68], food security status [69], dietary diversity [70,71]. In addition, the study lacked breastfeeding data [72,73] and more precise data about complementary feeding.

## 5. Conclusions

According to this research, the leading causes of stunting in children under five in East Nusa Tenggara province include the child’s age, the mother’s educational level, and the child’s place of residence. The highest stunting prevalence occurred in children aged 12–23 months (45.2%), with a stunting proportion by sex of 37.8% for males and 37.3% for females. Children aged 12–23 months are 2.08 times more likely to experience stunting than children aged 0–11 months. Children of mothers with low education are 1.57 times more likely to experience stunting than those of mothers with higher education. Moreover, mothers and children living in rural areas have 1.39 times higher risk of stunting than those living in urban areas. To reduce the incidence of stunting in the East Nusa Tenggara province, interventions that focus on increasing mothers’ knowledge of children under five about health, parenting, and nutrition are needed. Overall, it is necessary to increase the role of cadres in monitoring the growth and development of children under five in the Integrated Healthcare Center (Posyandu), especially in rural areas.

## Figures and Tables

**Table 1 ijerph-20-01640-t001:** Frequency distribution of respondents by characteristics (N = 1643).

Variable	*n*	%
Child’s age group (months)		
0–11	172	10.47
12–23	351	21.36
24–35	372	22.64
36–47	369	22.46
48–59	379	23.07
Child’s sex		
Male	838	51.0
Female	805	49.0
Supplementary feeding		
Yes	494	30.07
No	1149	69.93
Low birth weight infants		
Yes	0	0
No	1643	100
Child monitoring and development		
Yes	1345	81.86
No	298	18.14
Mother’s education		
Lower (≤JHS)	865	52.65
Higher (>SHS)	778	47.35
Mother’s occupation		
No	563	34.27
Work	1080	65.73
Place of residence		
Urban	469	28.54
Rural	1174	71.46
Access to healthcare facilities		
Easy	515	31.34
Difficult	1128	68.66
Stunting		
Yes	636	38.71
No	1007	61.29

**Table 2 ijerph-20-01640-t002:** Relationship between stunting and characteristics of respondents.

Characteristic	Stunting				OR	95% CI	*p*-Value
	Yes		No				
	*n*	%	*n*	%			
Child’s age group (months)							
0–11	35	23.8	137	76.2			
12–23	161	45.2	190	54.8	2.63	1.49–4.64	0.001 *
24–35	155	39.6	217	60.4	2.10	1.16–3.78	0.014 *
36–47	146	37.3	223	62.7	1.91	1.06–3.44	0.032 *
48–59	139	34.6	240	65.4	1.70	0.96–2.99	0.070
Child’s sex							
Male	337	37.8	501	62.2			
Female	299	37.3	506	62.7	1.023	0.77–1.35	0.874
Supplementary feeding							
Yes	201	40.3	293	59.7			
No	435	36.5	714	63.5	1.17	0.88–1.57	0.272
Mother’s education							
Higher (>SHS)	245	31.5	533	68.5			
Lower (≤JHS)	391	44.1	474	55.9	1.72	1.31–2.25	0.0001 *
Mother’s occupation							
Work	433	38.8	647	61.2			
No	203	35.7	360	64.3	0.88	0.67–1.15	0.343
Mother’s age							
≥20 Years	632	37.5	1000	62.5			
<20 Years	4	44.9	7	55.1	1.36	0.25–7.30	0.719
Child monitoring and development							
Yes	509	36.4	836	63.6			
No	127	42.5	171	57.5	1.29	0.96–1.75	0.095
Place of residence							
Urban	151	32.0	318	68.0			
Rural	485	41.0	689	59.0	1.47	1.1–1.99	0.009 *
Access to healthcare facilities							
Easy	182	34.5	333	65.5			
Difficult	454	39.4	674	60.6	1.23	0.92–1.65	0.166

* statistically significant related variables.

**Table 3 ijerph-20-01640-t003:** The final model of the related factors of stunting incidence in children.

Characteristic	B	SE	*p*-Value	OR	(95% CI)
Child’s age group (months)					
0–11 (Ref.)					
12–23	−0.510	0.317	0.108	2.78	1.52–5.09
24–35	0.511	0.199	0.011 *	2.08	1.12–3.86
36–47	0.221	0.202	0.276	1.93	1.03–3.60
48–59	0.146	0.202	0.469	1.67	0.89–3.10
Mother’s education					
Higher (>SHS)					
Lower (≤JHS)	0.448	0.145	0.002 *	1.57	1.18–2.08
Child’s monitoring and development					
Yes					
No	−0.290	0.163	0.075	1.34	0.97–1.84
Place of residence					
Urban					
Rural	−0.328	0.163	0.044 *	1.39	1.01–1.91

* statistically significant related variables.

## Data Availability

Not applicable.

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
