# Peer review of "Risk Factors for Stunting among Children under Five Years in the Province of East Nusa Tenggara (NTT), Indonesia"

_ijerph, 2023, doi:10.3390/ijerph20021640_

Round 1

Reviewer 1 Report

 The authors have done most the comments and the manuscript has been significantly improved and can be accepted in this form. 

Author Response

Thursday, January 5, 2023

Dear Editor-in-Chief

International Journal of Environmental Research and Public Health (IJERPH)

Cc. Review team

Hereby it is attested that the manuscript entitled “Risk Factors for Stunting Among Children Under Five Years in the Province of East Nusa Tenggara (NTT), Indonesia” submitted for publication in the International Journal of Environmental Research and Public Health (IJERPH). This article has been read and approved by all authors.

Manuscript revisions are made based on the questions from the review and answered point-by-point following the number of questions. This improvement manuscript can be considered for publication in the International Journal of Environmental Research and Public Health (IJERPH). We thank you for your time and cooperation

Best regards,

Dr. Gurendro Putro, MPH

National Research and Innovation Agency, Jakarta, Indonesia

Email:  gurendro.putro@brin.go.id

Reviewer 2 Report

The authors still conclude that “the leading causes of stunting in children under five in Province NTT were the child's age, the mother's educational level, and the child's place of residence” which is certainly correct, but this is not new. There is ample evidence that maternal education has important impact on child growth. Yet, it is not evident by which means maternal education influences child growth and it is premature to assume that “… education is related to the mother's knowledge of how to apply parenting patterns and the nutritional status of children.” Stunting is not a synonym of malnutrition (Scheffler et al. Stunting is not a synonym of malnutrition. Eur J Clin Nutr. 2020 Mar;74(3):377-386. doi: 10.1038/s41430-019-0439-4.). Growing evidence suggests that stunting independently relates to social disadvantage (Scheffler et al. Stunting as a Synonym of Social Disadvantage and Poor Parental Education. Int J Environ Res Public Health. 2021 Feb 2;18(3):1350. doi: 10.3390/ijerph18031350.), and that nutrition interventions have little or no effect on the growth of healthy stunted children.

This view, in fact, is corroborated in the present paper by the authors’ statement that “the variables of the child's gender, RECEIVING SUPPLEMENTAL FEEDING, mother's age, mother's occupation, monitoring and development of children and access to healthcare facilities were not significantly associated with the prevalence of stunting in children under five”. Further evidence is summarized in a recent meta-analysis indicating that “there was no evidence of an effect or unclear effect of nutrient supplementation of children on height-for-age for studies in the meta-analysis with low-certainty evidence, and inconclusive effect on length for studies reported in a narrative form with very low- to moderate-certainty evidence”, even in children living in urban slums (Goudet SM, Bogin BA, Madise NJ, Griffiths PL. Nutritional interventions for preventing stunting in children (birth to 59 months) living in urban slums in low- and middle-income countries (LMIC). Cochrane Database Syst Rev. 2019 Jun 17;6(6):CD011695. doi: 10.1002/14651858.CD011695.pub2. ).

The authors do not discuss this literature and prematurely and falsely conclude that “to reduce the incidence of stunting in the East Nusa Tenggara province, interventions that focus on increasing mothers' knowledge of children under five about health, parenting, and nutrition are needed.”

The study lacks information on feeding, respectively on body fat deposition e.g. on skinfold thickness, and particularly on observations on physical fitness and health. Being short per se, is not unhealthy.

The manuscript suffers from a very common problem: the discrepancy between the "official" knowledge on stunting and scientific evidence. Meanwhile, a large body of evidence questions the official view that being shorter than WHO is due to mal- oder undernutrition. This is NOT the case. I cited a few references. I already mentioned this problem in my first review. The authors simply ignore the recent scientific work and continue repeating official statements. Even in their own manuscript they state that "... the variables of the child's gender, receiving supplemental feeding, ....  were not significantly associated with the prevalence of stunting in children under five." But in spite of  their own findings, they finally conclude: "To reduce the incidence of stunting in the East Nusa Tenggara province, interventions that focus on increasing mothers' knowledge of children under five about health, parenting, and nutrition are needed. This is absurd. They do not need to increase mothers' knowledge about nutrition. It is not nutrition that makes children fail to grow. It is the complex social-economic-politial and emotional situation (see Patterns of Human Growth, new edition, by Barry Bogin) in many low- and middle income countries that differs from the situation in countries where children grow better. The problem is a political one - and this is one of the reasons, why the evidence about the lack of association between nutrition and child growth continues to be ignored, even in the scientific literature. This manuscript is another example..

Author Response

(The authors gave the same response as above.)

Reviewer 3 Report

Risk Factors for Stunting Among Children Under Five Years in 2 the Province of East Nusa Tenggara (NTT), Indonesia

This is a secondary analysis of the survey: Indonesia Basic Health Research, RISKESDAS, a large sample size and nationally representative sample. The aim of the authors was to study the factors associated with stunting in children under 5 years of age, from the East Nusa Tenggara Province.

The study is correctly planned and substantiated. In my opinion, some aspects to be improved:

Abstract:

·         It is structured and adequate.

·         2 wrong words: In line 5 ("dan") and in line 19 ("in-cidence").

·         In results, the direction of the associations should be indicated (lines 17 - 22).

Introduction:

·         It is properly written, justifies the importance of the study and ends with the objective.

Methodology:

·         The concept of the dependent variable must be clear (Line 110): “¿low? Children’s body height according to age”? It is well complemented with its definition at the end of the paragraph.

·         The independent variable "Supplementary feeding" is not defined in Methods, it should be clarified the question that is in the RISKESDAS survey, and what kind of supplements they received. This is an important issue because in many studies is generally associated to nutritional outcomes.

Results:

·         There is a phrase that has no sense: “There were no children with LBW who did not receive supplementary feeding (69.93%)” (Lines 167-168).

Looking at table 1, it seems that 69.93% of children were nor supplemented.

¿There were no children with LBW?

·         Specify the direction of the association: There was a significant relationship between the place of residence variable and the inci-dence of stunting in children (OR = 1.47, 95% CI: 1.1-1.99; p-value <0.05 (line 182-183).

·         Correct the word: “in-cidence”.

·         I suggest a figure with the distribution of this sample and the normal distribution of OMS length/age, could be more explicit the differences between them.

Discussion:

·         There is a phrase a little bit confuse about the influence of gender influence (Lines 205-207).

·         In line 216-217, the authors describe that “The results of the analysis showed that the lower the age group of children under five, the more likely they were to be at risk of stunting”. Iit must be changed, because afterwards, they point that those 12-23 months, had higher frequency of stunting than the 0-11 group (Lines 216-218).

·         After lines 230-233, it could be more precise that insufficient supplemental feeding produce insufficient energy, protein and micronutrient intake: all together impact growth.

·         In the paragraph that the authors discuss about the lack of association of supplementary feeding and stunting is important to consider the limitation of the question that the survey address it. For that reason is important to define it in the Methods (type of feeding, frequency, quantity, cultural adequacy?). (lines 236-246)

·         The absence of association of stunting and monitoring could be because reversal causality?

·         Limitations: to not have breastfeeding data and more precise data about complementary feeding.

Conclusions:

·         Well done, with a good end projection.

Author Response

(The authors gave the same response as above.)

Round 2

Reviewer 2 Report

The manuscript has improved. Yet, the authors still do not dare to properly present their findings, and to draw the respective conclusions.

It is quite OK to start the manuscript with the conventional wisdom that “ … chronic nutritional problems are caused by a lack of nutrient intake in the long term, resulting in unfulfilled nutritional needs, causing stunting” and to present stunting rates in Indonesia and East Nusa Tenggara. But already in the introduction, the authors should prepare the reader for their findings. Table 2 clearly indicates that supplementary feeding has no effect on the children of their study. In fact, the percentage of stunted children is HIGHER in the supplemented group than in the controls. I think there is no better argument AGAINST the conventional wisdom. The stunted children of this study did obviously NOT suffer from undernutrition because they showed NO catch-up growth after nutritional supplementation.

The lack of association between being shorter than WHO standard and nutrition is one of the main messages of this manuscript and needs to be properly presented. It is in line with the studies of Scheffler et al. and with numerous other recent studies that FAIL to provide evidence for any causal relation between nutrition and body height in otherwise healthy children from middle- and low-income countries.

In light of their findings, the authors should refrain from repeating that “… providing additional food is one of the health efforts to monitor the growth and development of infants and children so that they can prevent and treat malnutrition as early as possible”. STUNTING IS NOT A SYNONYM OF MALNUTRTION. Being shorter than WHO standards is caused by social, educational, and certainly also by political factors. This appears true also for the present manuscript and needs to be stated.

Short remark: The authors should check the term “gender”. The distinction between sex and gender differentiates a person's sex (the anatomy of an individual's reproductive system, and secondary sex characteristics) from that person's gender, which can refer to either social roles based on the sex of the person (gender role) or personal identification of one's own gender based on an internal awareness (gender identity).

Author Response

Tuesday, January 10, 2023

Dear Editor-in-Chief

International Journal of Environmental Research and Public Health (IJERPH)

Cc. Reviewer Team

Hereby it is attested that the manuscript entitled “Risk Factors for Stunting Among Children Under Five Years in the Province of East Nusa Tenggara (NTT), Indonesia” submitted for publication in the International Journal of Environmental Research and Public Health (IJERPH). This article has been read and approved by all authors.

Manuscript revisions are made based on the questions from the review and answered point-by-point following the number of questions. This improvement manuscript can be considered for publication in the International Journal of Environmental Research and Public Health (IJERPH). We thank you for your time and cooperation

Best regards,

Dr. Gurendro Putro, MPH

National Research and Innovation Agency, Jakarta, Indonesia

Email:  gurendro.putro@brin.go.id
